# Development of a new genotype–phenotype linked antibody screening system

**Takashi Watanabe[1]\*, Hikaru Hata[2], Yoshiki Mochizuki[1], Fumie Yokoyama[1], Tomoko Hasegawa[1], Naveen Kumar[3], Tomohiro Kurosaki[2,4], Osamu Ohara[1,5], Hidehiro Fukuyama[2,6,7]**

[1]Laboratory for Integrative Genomics, RIKEN Center for Integrative Medical Sciences, Yokohama, Japan; [2]Laboratory for Lymphocyte Differentiation, RIKEN Center for Integrative Medical Sciences, Yokohama, Japan; [3]Laboratory for Integrated Bioinformatics, RIKEN Center for Integrative Medical Sciences, Yokohama, Japan; [4]Laboratory of Lymphocyte Differentiation, WPI Immunology Frontier Research Center, Osaka University, Osaka, Japan; [5]Department of Applied Genomics, Kazusa DNA Research Institute, Chiba, Japan; [6]Division of Immunology, Near-InfraRed Photo-Immunotherapy Research Institute, Kansai Medical University, Osaka, Japan; [7]INSERM EST, Strasbourg Cedex, France

**\*For correspondence:**
twatanabe@riken.jp

**Competing interest:** The authors declare that no competing interests exist.

## eLife Assessment

The **useful** studies described here are broadly applicable to all antibody discovery subfields, even though they are not a significant improvement over published methods. The findings are **incomplete** with respect to the methodology, since details that are crucial in order to repeat the experiments are lacking (such as a timestamp). They also do not take into account multiple recent papers that have tested similar strategies. These studies will be of interest to a specialized audience working on generating antibodies to infectious agents.

**Abstract** Antibodies are powerful tools for the therapy and diagnosis of various diseases. In addition to conventional hybridoma-based screening, recombinant antibody-based screening has become a common choice; however, its application is hampered by two factors: (1) screening starts after Ig gene cloning and recombinant antibody production only, and (2) the antibody is composed of paired chains, heavy and light, commonly expressed by two independent expression vectors. Here, we introduce a method for the rapid screening of recombinant monoclonal antibodies by establishing a Golden Gate-based dual-expression vector and in-vivo expression of membrane-bound antibodies. Using this system, we demonstrate the rapid isolation of influenza cross-reactive antibodies with high affinity from immunized mice within 7 days. This system is particularly useful for isolating therapeutic or diagnostic antibodies, for example during foreseen pandemics.

## Introduction

Technical improvements in the rapid isolation of monoclonal antibodies (mAbs) are critical for the diagnostic and therapeutic development of such antibodies. This need became apparent to the general public during the COVID-19 pandemic. Antibodies isolated from convalescent COVID-19 patients are valuable resources for the development of mAb therapeutics and rapid diagnostic tools.

Human antibodies can be administered to other patients without the need to humanize in an emergency and with a slight modification of Fc receptor binding (*Balsitis et al., 2010*) to increase the half-life or reduce the risk of antibody-dependent enhancement. Although production costs remain high, the efficiency of generating recombinant antibodies has dramatically improved with the use of mammalian cell lines, and the application of therapeutic antibodies is likely to expand in the near future (*Fang et al., 2017*).

The gold standard method for isolating mAbs is the hybridoma technology developed by Kohler and Milstein in 1975 (*Köhler and Milstein, 1975*), in which antibody-producing B cells are fused with immortal B cells, called myelomas, to produce long-lasting antibody-producing B cells. In addition to hybridoma technology, new methods have been developed to increase mAb screening efficiency. These include direct immortalization of B cells by gene reprogramming using the Epstein–Barr virus (*Traggiai et al., 2004*) or retrovirus-mediated gene transfer (*Kwakkenbos et al., 2010*), cloning of variable region-encoding genes by single-cell PCR (*Wrammert et al., 2008*; *Meijer et al., 2006*), single-cell culture screening (*Kuraoka et al., 2016*), and in vitro screening of recombinant antibody libraries (*Clackson et al., 1991*; *Feldhaus et al., 2003*; *Harvey et al., 2004*; *Schaffitzel et al., 1999*; *Mazor et al., 2007*). Some of these methods have successfully yielded high-affinity antibodies in various formats, including single variable domain on a heavy chain or single-chain fragment variable antibodies. Although hybridoma technology is highly reliable for the isolation of valuable antibodies from animal models, such as mice, rats, and hamsters, it is time-consuming and requires technical skill. The development of a new fusion partner cell line, SPYMEG, has enabled the production of human hybridomas and opened a new direction for the isolation of human mAbs (*Kubota-Koketsu et al., 2009*). Although many of these new technologies have increased the throughput for mAb isolation, the procedure still requires significant resources and time.

To dramatically enhance the efficiency of mAb isolation, we used next-generation sequencing (NGS) technology, which has revolutionized the sequencing of immunoglobulin (Ig) variable-region genes. For instance, tens of thousands of Ig genes specific to certain antigens can be identified by combining droplet-based single-cell isolation with DNA barcode antigen technology, followed by NGS (*Setliff et al., 2019*). Although Ig genes can be sequenced at high throughput, there is no method for screening antibodies in a high-throughput format that is compatible with NGS technology.

In this study, we developed a new functional screening method that is compatible with NGS to rapidly identify antigen-specific clones. We first generated an Ig dual-expression vector using Golden Gate Cloning (*Kirchmaier et al., 2013*), which enabled the linkage of heavy-chain variable and light-chain variable DNA fragments obtained from a single-sorted B cell, followed by the expression of membrane-bound Ig. This single-step procedure enabled the enrichment of antigen-specific, high-affinity Igs by flow cytometry, which is significantly faster than conventional cloning-based methods that require sequential steps. To demonstrate the efficiency of our new method, we screened for potent broadly reactive antibodies (*Okuno et al., 1993*; *Ekiert et al., 2012*; *Impagliazzo et al., 2015*) against the influenza virus in an experimental mouse model using our technology. Broad reactivity antibodies allow for the development of influenza vaccines that are effective across seasonal variations in influenza strains, which can only be accomplished by screening a large number of candidate antibodies. We first raised cross-reactive B cells against various hemagglutinin (HA) antigens from the influenza virus by sequential immunization with heterotypic HA antigens from group 1 influenza. Using this technology, we obtained several mAbs that bind to other group 1 HA antigens, and even to group 2 HA antigens, of the influenza virus. Our technology can also be applied to human antibody screening and represents a new line of mAb screening that accelerates the isolation of therapeutic and diagnostic mAbs.

## Results

### Rapid membrane-bound dual Ig expression screening system

Cloning-based Ig screening involves several steps: cloning two Ig gene fragments that encode heavy and light chains independently from a single cell, co-expressing these Ig chains, and purifying individual recombinant antibodies. These steps are labor intensive. We established a new antibody screening system for the isolation of valuable mAbs (details in the methods). Eventually, the antigen-binding Ig

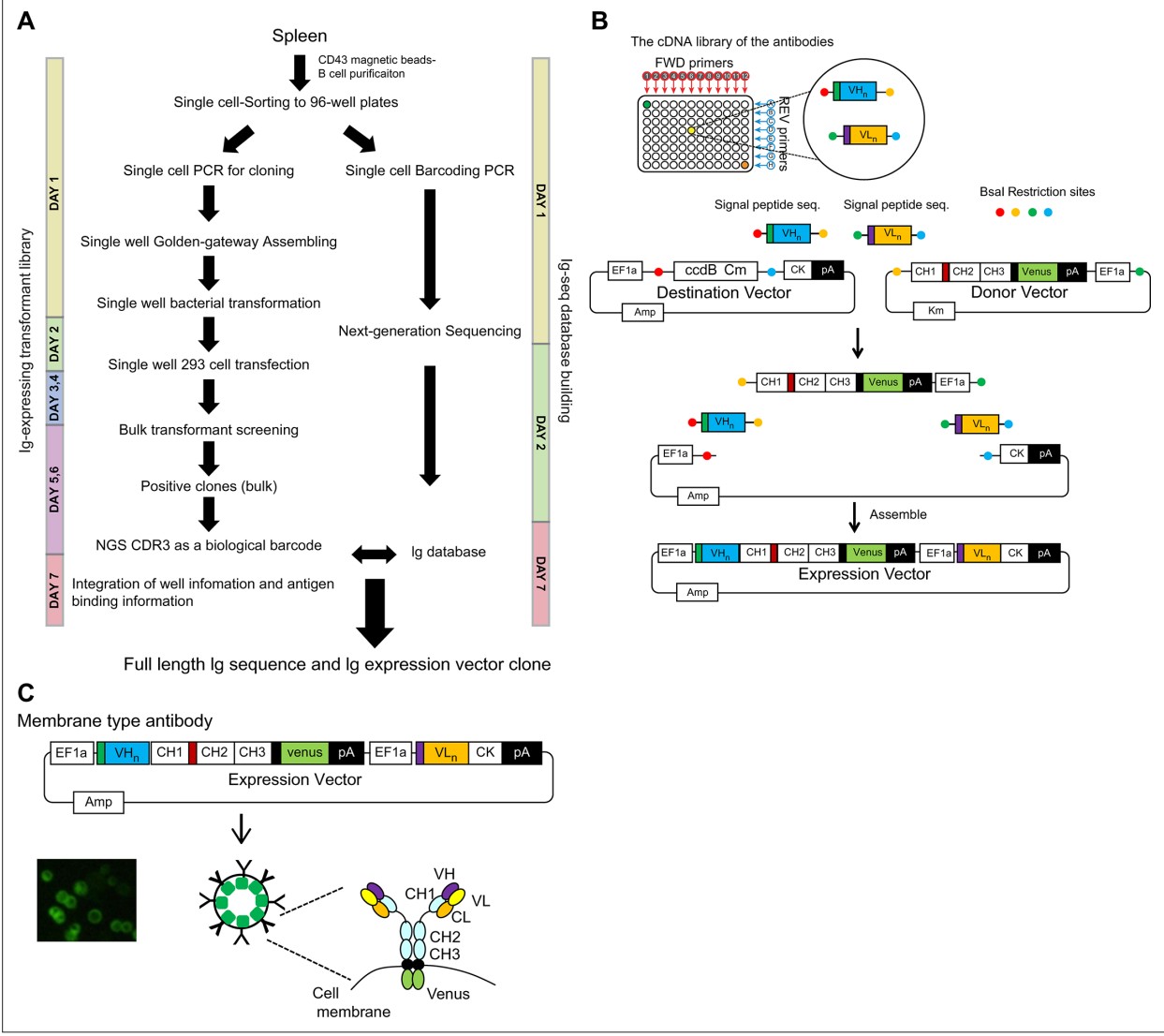

**Figure 1.** System to link the antigen-binding feature with the Ig repertoire genetic information. (**A**) Single cell-sorted B cells were subjected to two processes: Ig-seq database building and Ig-expressing transformant library preparation. Antigen-binding Ig transformants were collected by sorting in bulk and the sequences of the unique CDR3 regions and clones of interest were identified by referring to the Ig-seq database. The duration of each steps are indicated. (**B**) To express both Ig heavy and kappa/lambda chains in a single-expression vector, we generated a dual-expression vector. Four gene fragments were assembled by the Golden Gate method using the BsaI restriction enzyme. They included: (1) the destination vector containing the Ig kappa constant gene and the ccdB Chloramphenicol cassette for negative selection, (2) fragments containing the Ig heavy constant gene fused to *Venus* derived from the donor vector, (3) the Ig heavy variable gene fragment, and (4) the Ig light variable gene fragment. (**C**) Individually purified plasmids were transiently transfected to the floating human FreeStyle 293 cell line. Igs were expressed on the cell surface in 2 days, and expression levels were confirmed and normalized by *Venus* expression, since the Ig heavy chain was fused to *Venus* at the cytoplasmic domain tail.

The online version of this article includes the following figure supplement(s) for figure 1:

**Figure supplement 1.** The Ig repertoires are visualized using the in-house software.

**Figure supplement 2.** Progression flowchart.

**Figure supplement 3.** Sorting strategy.

transformants were collected by sorting in a bulk fashion, and the unique CDR3 region and clones of interest were identified using the Ig-seq database (*Figure 1A*).

We generated a dual-expression vector to express both Ig heavy and kappa/lambda in a single-expression vector (*Figure 1B*). These plasmids were individually transfected transiently to the floating human FreeStyle 293 cell line. Igs were expressed on the cell surface within 2 days, and their expression could be confirmed and normalized by *Venus* (*Nagai et al., 2002*) expression since the Ig heavy

chain is fused with *Venus* at the cytoplasmic domain tail (*Figure 1C*). The advantage of this system is the rapid enrichment of clones of interest through bulk screening. In the current single-cell-based cloning/screening, plasmid DNA extracted from the collected antigen-binding bulk transformants is sequenced for the heavy chain CDR3 region. The CDR3 region is unique and can be used for the identification of a clone (*D'Angelo et al., 2018*; *Xu and Davis, 2000*). This membrane-bound Ig expression system links antigen-binding features and genetic information of the Ig repertoire.

## Building the Ig-seq database and analysis

As a model experiment, we obtained broadly reactive antibodies against influenza viruses using multiple HA probes. We prepared two HA proteins as probes: A/Puerto Rico/8/1934 (H1N1), designated as PR8, and A/Okuda/1957 (H2N2), designated as H2. A total of 374 IgG1$^+$ B cells of either PR8$^+$ (204 cells), H2$^+$ (99 cells), or PR8$^+$H2$^+$ (71 cells) were collected in a single-cell fashion. We obtained sequences of 284 independent clones. The success rate of cloning the paired Ig fragments was 75.9%. An overview of heavy chain V-D-J and light chain V-J usage and repertoire clonality is shown in *Figure 1—figure supplement 1A and B*. Mutation rates and CDR3 lengths of the heavy chains of the three cell populations (PR8$^+$, H2$^+$, and PR8$^+$H2$^+$) were comparable (*Figure 1—figure supplement 1C and D*). These results indicate that broadly reactive antibodies do not require unique genetic traces to obtain breadth in our experimental setup.

## Isolation of H2- and H1-reactive B cells

On day 2 after transfection of individual 284 clones, the transformants were mixed and stained with HA probes (*Figure 2A*). As shown in *Figure 2B*, we selected three prominent populations, H1$^+$H2$^+$ (cross), H2$^+$ (H2), and H1$^+$ (H1), and collected them in bulk. In contrast to bacterial transfection, mammalian cell lines, such as FreeStyle 293 cells, can contain multiple plasmids in a cell and ectopically express multiple proteins. Therefore, we decided to mix the transformants instead of plasmids. We sorted a total of 2981 cells, classified into three strong binder populations, and performed sequencing (*Figure 2B*). These bulk Ig-seq data are referred to as Ig-seq data, as described above. A total of 190 clones from the three populations were verified for their sequences and HA binding by flow cytometry analysis, as shown in *Figure 2C*; 110 'H1$^+$', 67 'H2$^+$', and 13 'cross' clones were successfully isolated. In parallel, all the transformants were examined individually for comparison with the bulk method. One hundred thirty-eight transformants were verified for their binding to HAs by flow cytometry. Of these, 81 'H1$^+$', 48 'H2$^+$', and 9 'cross' clones were successfully isolated (*Figure 2D*). Eight of the 13 'cross' clones obtained in the bulk examination overlapped with 'cross' clones obtained in the individual examination (*Figure 2D and E*); the binding of one cross clone (D11p4) from the individual examination was so weak that it was not included in the gates sorted in bulk (*Figure 2B*). Comparison of the bulk and the individual examinations demonstrated that the rapid bulk method is sufficient for selecting clones of interest.

## Characterization of 'cross' clones for broad reactivity against the influenza virus

We decided to further characterize eight common clones between the individual and bulk examinations and one short-ranged clone, D11p4. By generating a secretory form of mAbs for these nine 'cross' clones from the individual examination, we characterized their affinity and broadness using six different HA antigens from strains A/Okuda/1957 (H2N2) named for H2, A/Puerto Rico/8/1934 (H1N1) for PR8, A/California/2009 (X-179A) [H1N1] Pdm09 for Cal, A/Texas/50/2012 (X-223) [H3N2] for H3, A/Egypt/N03072/2010(H5N1) for H5, and A/Brisbane/59/2007 [H1N1] (*Impagliazzo et al., 2015*) for Stem. We determined the affinities ($K_d$) of the antibodies for the six HA antigens using surface plasmon resonance analysis (*Table 1* and *Figure 2—figure supplement 1*), which ranged from 500 to 100 nM. With the highest affinity among them, the C10p2 antibody bound to Cal at $K_d \simeq 5.66 \times 10^{-10}$ (M). Surprisingly, the antibodies showed a broader spectrum of HA binding among the influenza strains. Seven of the nine mAbs bound to HA from the highly pathogenic avian influenza strain H5N1. Furthermore, six antibodies that bound to H3 were categorized into group 2 (*Gamblin and Skehel, 2010*). In addition, we observed that two of the broadly reactive antibodies recognized the stem region of the HA of A/Brisbane/59/2007. A possible scenario for other stem-negative antibodies is that they may either bind to the stem region of other strains or to the head region of HA. We

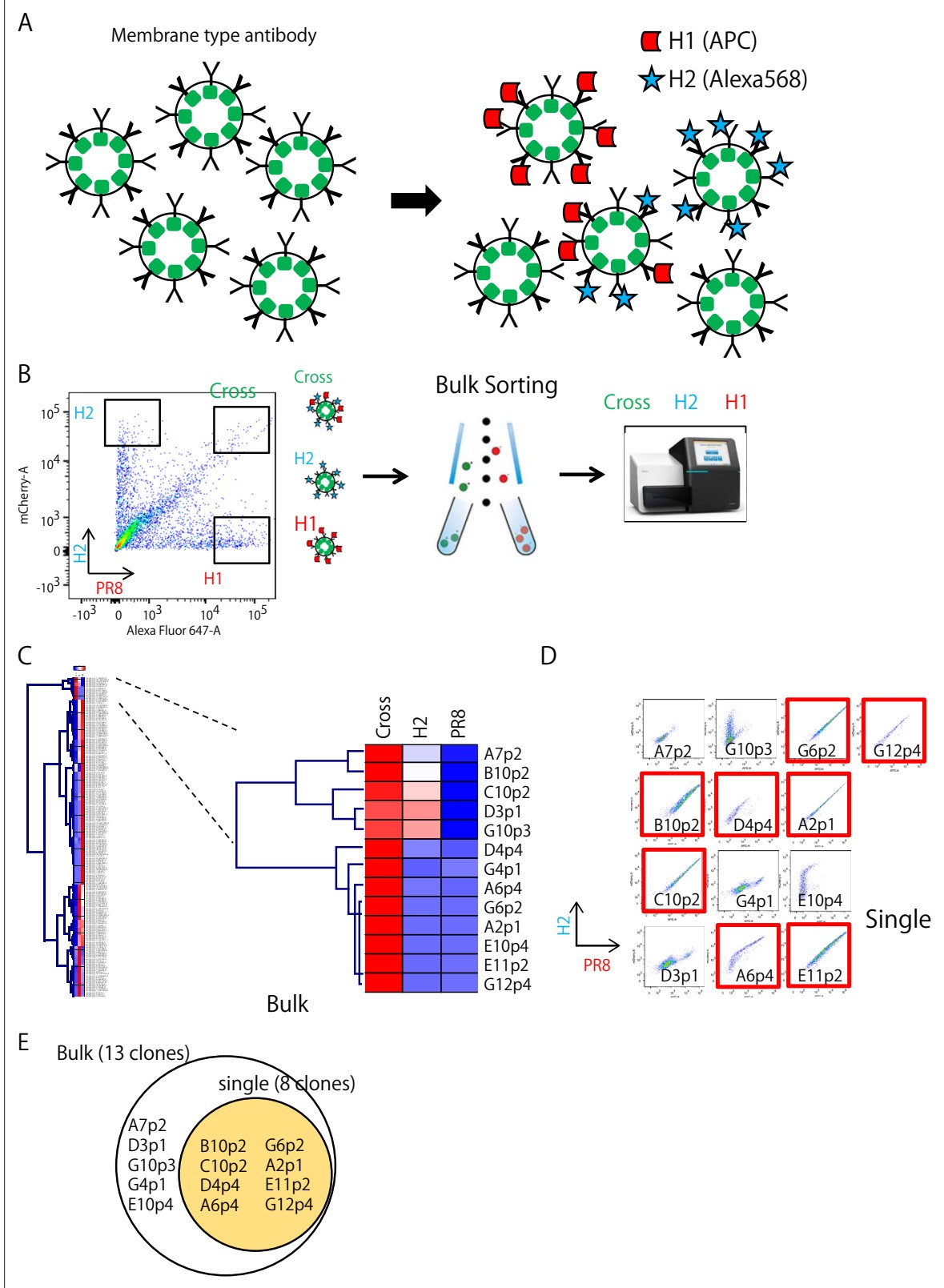

**Figure 2.** Isolation of H2 and H1-reactive B cells. (**A**) Mixture of Ig-expressing transformant libraries stained with HA probes. (**B**) Three strong HA-binding populations, H1+H2+ (cross), H2+ (H2), and H1+ (H1), were sorted and collected in a bulk fashion. Then, these three collected populations were and sequenced. (**C**) The bulk Ig-seq data were referred to as the single-cell Ig-seq data. Heatmap indicates the appearance rate of individual clone in each box shown in B (H1, H2, or Cross) among sorted 2981 cells of mixture of 190 clone transformants. H1+H2+ (cross) clones are shown on the right

*Figure 2 continued on next page*

*Figure 2 continued*

columns. High to low appearance rate reflects red to blue color. (**D**) The individual flow cytometry profile of 13 'Cross' transformants obtained by the bulk examination are shown and red labeled ones are also found by the individual examination(red). (**E**) A summary of the results in comparison of the bulk and individual examination methods.

The online version of this article includes the following figure supplement(s) for figure 2:

**Figure supplement 1.** SPR profiles of kinetics analysis of nine clones, A6p4, B10p2, C10p2, A2p1, D11p4, E11p2, G6p2, D4p4, and G12p4 using a BIAcore 3000 machine (GE Healthcare).

**Figure supplement 2.** Two hundred and eighty-four clones are plotted in a two-dimensional map of phylogeny trees for heavy and light chain genes.

**Figure supplement 3.** The affinity of antigen-antibody binding (in this case, probe and membrane-bound antibody expressing cell) can be inferred from the population shift (flow cytometry analysis).

tested whether some of these antibodies shared features with classic broadly neutralizing antibodies. C179, the first broadly neutralizing mouse antibody isolated in 1993 (*Okuno et al., 1993*), reacts with the stem region of HAs from group 1 influenza virus strains (*Gamblin and Skehel, 2010*). C179 binds to HAs from group 1 influenza viruses, including H1N1, H2N2, and H5N1. A competition assay for the binding of H2 and PR8 revealed that A6p4 competes with C179 (*Figure 3*). In addition, our two-dimensional phylogeny map indicated that A6p4 was unique to all clones (*Figure 2—figure supplement 2*). Note that we used NSP2 as a negative control antibody, which binds specifically to the HA from A/California/2009 (X-179A) [H1N1] Pdm09 (*Adachi et al., 2015*). These results demonstrate that we were able to isolate a broadly reactive mAb with an affinity comparable to that of a classic broadly reactive mAb (C179) using our technology.

## Discussion

The discovery of therapeutic antibodies as well as their demand are increasing (*Kaplon et al., 2023*). Efficiently isolating one biologically significant antibody from the $10^{13}$ antibody repertoire is challenging. The hybridoma method has long been used for this purpose and remains reliable. However, the fusion rate of B and myeloma cells is <0.001% in this method; therefore, the number of hybridomas obtained from one individual B cell is limited to several hundred clones, limiting its use in the

**Table 1.** Affinities ($K_d$) of the antibodies to six HA antigens from strains A/Okuda/1957(H2N2), termed H2, A/Puerto Rico/8/1934(H1N1), termed PR8, A/California/2009 (X-179A) [H1N1] Pdm09, termed Cal, A/Texas/50/2012 (X-223) [H3N2], termed H3, A/Egypt/N03072/2010(H5N1), termed H5, and A/Brisbane/59/2007 (H1N1), termed Stem, were determined using surface plasmon resonance analysis.

H2, PR8, Cal, and H5 belong to group 1, and H3 belongs to group 2. High (red), Middle (orange), and Low (blue).

| Group | 1 | 1 | 1 | 2 | 1 | 1 |
|---|---|---|---|---|---|---|
| | H2 | PR8 | Cal | H3 | H5 | Stem |
| A6p4 | 4.51E-08 | 1.41E-06 | 1.14E-05 | | 2.91E-05 | 1.59E-08 |
| B10p2 | 6.75E-09 | 2.3E-09 | 3.29E-09 | | | |
| C10p2 | 1.66E-09 | 1.29E-09 | 5.66E-10 | | | 4.26E-08 |
| A2p1 | 2.49E-06 | 1.42E-07 | 1.14E-05 | 1.25E-04 | 1.91E-05 | |
| D11p4 | 7.93E-07 | 1.72E-06 | 1.04E-06 | 1.36E-05 | 1.05E-11* | |
| E11p2 | 4.4E-08 | 1.86E-08 | 1.67E-06 | 2.92E-07 | 2.37E-07 | |
| G6p2 | 3.51E-08 | 3.53E-08 | 1.44E-08 | 1.78E-05 | 3.05E-08 | |
| D4p4 | 1.1E-05 | 1.78E-05 | 6.4E-05 | 4.38E-05 | 2.47E-05 | |
| G12p4 | 1.45E-07 | 1.52E-07 | 5.7E-07 | 9.44E-07 | 1.58E-07 | |

Surface plasmon resonance (Biacore) KD (M).

*The affinity of D11p4 for H5 was low and inaccurate (**Figure 2—figure supplement 1**).

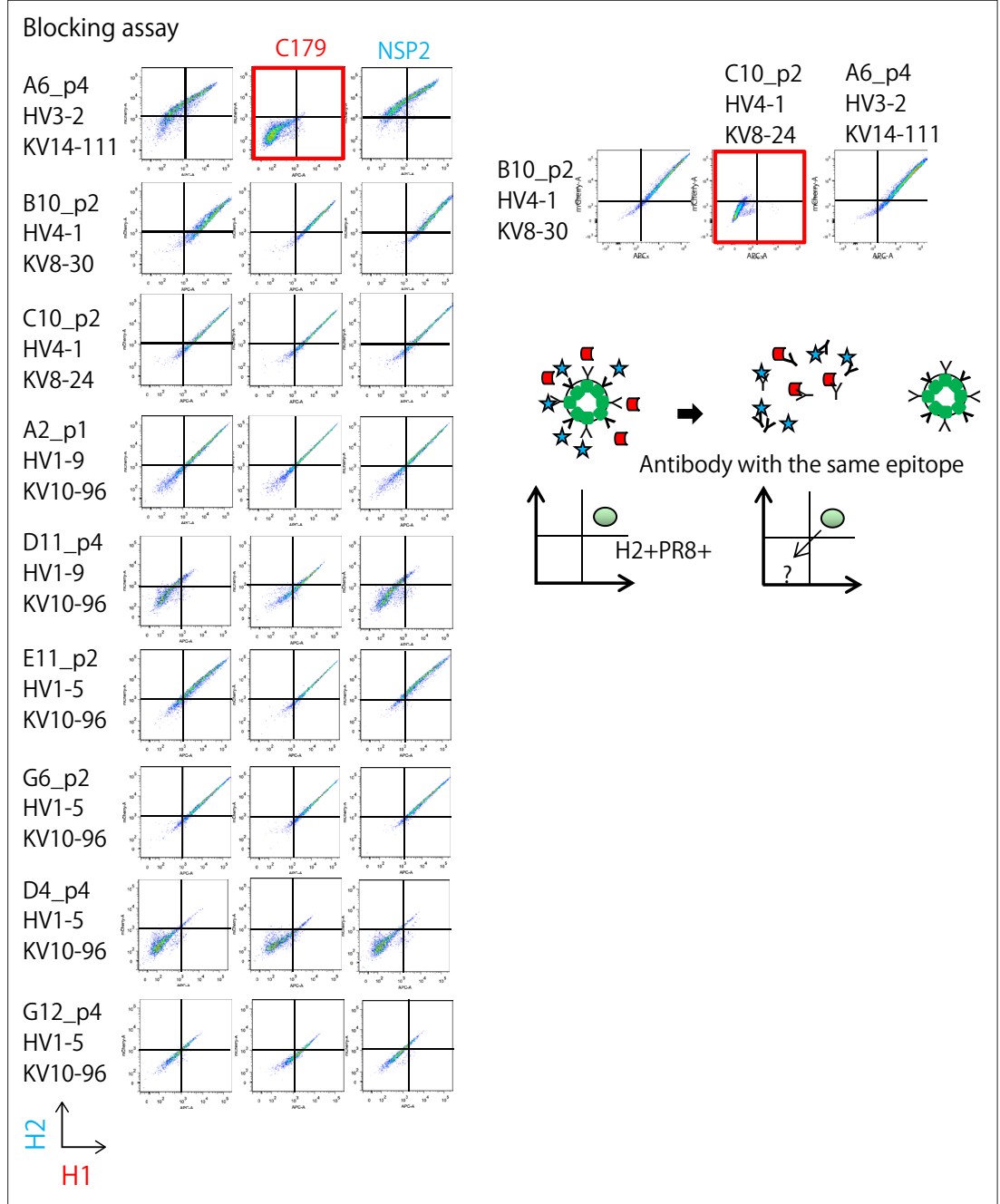

**Figure 3.** Competition assay of 'cross' Ig-transformants using C179, a broadly-reactive mAb binding to the Stem region of HA, and NSP2 (Narita strain-specific mAb), a strain-specific mAb that binds to the head region of HA. On the right panel, two stem-reactive clones (see *Table 1*) were examined reciprocally with combination of transformant and recombinant antibodies.

production of mAbs. As an alternative choice, cloning-based antibody screening has gained popularity, especially for human antibody discovery, because of the lack of a human myeloma cell line similar to that used for murine cell fusion partners. The SPYMEG cell line can generate antigen-specific antibody-producing cells by fusing the murine SP2/0 myeloma cell line and human megakaryoblastic leukemia cell line MEG-01 with human PBMCs (*Kubota-Koketsu et al., 2009*; *Soni et al., 2018*). In our study, we sorted IgG $1 +$ B cells. Most of those cross-reactive B cells were memory B cells (*Figure 1—figure supplement 2*). As shown in *Figure 1—figure supplement 3*, the cloning efficiency exceeded 75%. Our functional screening by selecting strong binders allowed us to narrow down the number of clones to 13 cross-reactive B cells. One of the clones had a strong affinity of $K_d \simeq 5.66\times10^{-10}$

(M). Notably, the generation of antibodies that target regions other than HA cannot be ruled out since the immunization antigen and the detected antigen were the same. However, as shown in *Table 1*, the cross-reactive antibodies obtained in this study exhibited characteristic binding abilities to each of the six types of HAs. If these were antibodies recognizing His-tag, they would bind to all six types of HA, indicating that these cross-reactive antibodies were no His-specific clones.

In addition to gene expression profiling, NGS facilitates antibody repertoire analysis. However, despite breakthroughs in sequencing technology, the process of linking antibody function and its genes is still long and labor intensive. To overcome this issue, we developed a new antibody screening system to directly link antigen–antibody binding (as a function) with a gene encoding the antibody by expressing its membrane-bound form. Among conventional cloning-based mAb isolation methods, this system demonstrates several advantages. (1) Membrane Ig expression can link the antigen-binding feature of membrane-expressed Ig, which can be linked to Ig DNA sequence information using our plasmid construct. Similar to the panning procedure used in phage display (*Petropoulos, 2012*), antigen-binding cells carry the plasmid encoding Ig genes through surface Ig so that the selection process in bulk format enriches the relevant plasmids for use in further experiments. (2) The dual Ig expression vector links heavy- and light-chain genes, which reduces the plasmid preparation time and stock by half. (3) Highly reliable Golden Gate Cloning technology using type IIs restriction enzymes can readily generate plasmid clones. This reduced the time required to generate an Ig plasmid library. (4) The population profile, defined by the fluorescence intensity during flow cytometry, directly reflected the affinity of a clone (*Figure 2—figure supplement 3*). Lima et al. developed a dual Ig expression vector, similar to the one we report here, to produce secretary antibodies and succeeded in obtaining functional monoclonal antibodies from SARS-CoV-2 infected individuals (*Lima et al., 2022*). Collectively, our technology streamlines the isolation of mAbs for therapy and diagnosis.

## Conclusion

Our findings indicate that the developed antibody presentation system facilitates antibody functional analysis and is well suited for the discovery of antibodies important for infectious diseases when combined with conventional NGS-based antibody repertoire analysis. Compared with droplet-based experimental systems, well-based systems are limited in the number of cells they can process. Furthermore, experiments involving infectious bacteria and viruses have imposed limitations on human experimentation. To solve these problems, the automation of experiments will become important in the future. By combining our screening system with robotic automation of experiments, it will be possible to obtain useful mAbs for various diseases quickly and in large quantities, which has broad implications for the development of vaccines against various diseases.

## Materials and methods

### Key resources table

| Reagent type (species) or resource | Designation | Source or reference | Identifiers | Additional information |
|---|---|---|---|---|
| Gene (*Aequorea coerulescens*) | venus | RIKEN BRC; *Nagai et al., 2002* | | |
| Strain, strain background (*Escherichia coli*) | DH5α | Thermo Fisher Scientific | EC0112 | competent cells |
| Genetic reagent (F' Episome) | ccdB | Thermo Fisher Scientific | V79020 | pcDNA3.1 (+) Mammalian Expression Vector |
| Cell line (*Homo sapiens*) | FreeStyle 293 | Thermo Fisher Scientific | R79007 | |
| Transfected construct (*M. musculus*) | Antibody expression vector | This paper | | Plasmid construct to transfect and express the antibody. |
| Biological sample (*M. musculus*) | Mouse splenocytes | This paper | CLEA Japan, Inc. | |
| Antibody | Anti-CD43 MicroBeads (Mouse monoclonal) | Miltenyi Biotec, Inc. | 130-049-801 | Add 10 µL of Anti-CD43 (Ly-48) MicroBeads (mouse) per $10^7$ total cells (1:1000 dilution) |

*Continued on next page*

*Continued*

| Reagent type (species) or resource | Designation | Source or reference | Identifiers | Additional information |
|---|---|---|---|---|
| Recombinant DNA reagent | pcDNA3.4-mIgG1 (plasmid) | This paper | | To obtain a large amount of secretory antibodies |
| Recombinant DNA reagent | pcDNA3.4-kappa (plasmid) | This paper | | To obtain a large amount of secretory antibodies |
| Sequence-based reagent | BsaI_IL6sp_L | This paper | PCR primers | CTAGGGTCTCAAGCAGATG AACTCCTTCTCCACAAGCG |
| Sequence-based reagent | mC_G_new2_BsaI | This paper | PCR primers | TCCTAGGTCTCCCACACACA GGGGCCAGTGGATAGAC |
| Peptide, recombinant protein | Anti-HA antibodies | This paper | | Nine cross reactive antibodies |
| Commercial assay or kit | BsaI-HFv2 | New England Biolabs | NEB #R3733 | |
| Commercial assay or kit | T4 DNA ligase | New England Biolabs | M0202T | |
| Chemical compound, drug | AddaVax | InvivoGen | vac-adx-10 | adjuvant |
| Software, algorithm | BONSCI | in-house software | | Ig database construction and visualization of the Ig repertoire |
| Other | CM5 sensor chip | GE Healthcare Technologies | BR100530 | 3 sensor chips |

## Protein purification of influenza HAs (H1 and H2) as antigens

A mixture of 25.7 µg of HA, comprising H1 strain A/Puerto Rico/8/1934 (PR8), H1 strain A/California/7/2009 (X-179A), H2 strain A/Okuda/1957 (H2), H3 strain A/Texas/50/2012 (X-223), or H5 strain A/Egypt/[N03072/2010], 1.3 µg of NA (Strain-matched), and 3 µg of BirA-expressing plasmids were transfected into Expi293 cells using the Expifectamine 293 transfection kit (Thermo Fisher Scientific, Waltham, MA, USA) according to the manufacturer's instructions. The cells were then cultured in Expi293 Expression Medium (Thermo Fisher Scientific) supplemented with D-biotin at a final concentration of 100 µM for in vivo BirA biotinylation in a humidified incubator containing 8% $CO_2$ at 37 °C and 125 rpm. If biotinylation was not necessary, transfection was performed without BirA, and cell culture was carried out without D-biotin. Culture supernatants were harvested 3 days post-transfection and filtered through a 0.45 µm filter (Merck Millipore, Burlington, MA, USA). All proteins were purified with a Talon metal affinity resin (Takara Bio USA, Mountain View, CA, USA) and dialyzed against pyrogen-free PBS. The oligomeric state and purity were determined at 25 °C by size-exclusion chromatography on a Superdex 200 10/300 GL column (GE Healthcare Technologies, Chicago, IL).

## HA (H1 and H2) protein immunization

Two BALB/c mice were intraperitoneally immunized sequentially, 2 weeks apart, with 15 µg of H1 (PR8) HA, followed by 15 µg of H2 HA protein as an antigen and supplemented with AddaVax adjuvant (InvivoGen, San Diego, CA, USA). Two weeks after the second immunization, the mice were sacrificed, and their splenocytes were used for the experiment. BALB/c mice were purchased from CLEA Japan (CLEA Japan, Inc, Tokyo, Japan). All animal experiments were performed using protocols approved by the Institutional Animal Care and Use Committee (IACUC) of the RIKEN Yokohama Branch (Project title: Immunological memory and vaccine development/Approval No.: 2019–001).

## Fluorescence-activated cell sorting

Using ACK (Ammonium-Chloride-Potassium) lysing buffer-treated splenocytes, CD43-negative B cells were collected using AutoMACS (Miltenyi Biotec, Inc, Bergisch Gladbach, North Rhine-Westphalia, Germany) and stained with IgG1 (Bv510), non-biotinylated His-tagged purified recombinant His-tagged H1 (PR8) protein, and CD38 (PE-Cy7) on ice for 30 min. After washing three times, the cell suspension was incubated with Alexa Fluor 488-labeled anti-His antibodies on ice for 30 min. After washing three times, biotinylated purified recombinant H2 protein was incubated on ice for 30 min. Furthermore after washing three times, the cell suspension was incubated with Brilliant Violet

421-conjugated streptavidin on ice for 3 min. We used 7-AAD for excluding dead cells. Next, the B cells were subjected to single-cell sorting using a BD FACSAria III (BD Biosciences, Franklin Lakes, NJ, USA) and collected into 96-well plates pre-loaded with lysis buffer. The single cells in the lysis buffer were immediately snap-frozen on powdered dry ice and stored at −80 °C.

## Amplification of a paired B-cell repertoire amplicon from a single cell

Single cells were dropped into pre-loaded plates containing 4 µL/well of lysis buffer (2 U/µL RNase inhibitor, 0.5×PBS, 10 mM DTT) and 1 µL of 10 µM Oligo(dT)18 Primer. Cell lysates were incubated at 70 °C for 90 s, followed by incubation at 35 °C for 15 s, and then placed on ice for 2 min. For first-strand cDNA synthesis, 5 µL of RT mix (2×PCR buffer, 5 mM DTT, 2 mM dNTP Mix, 10 U RNase inhibitor, 100 U SuperScript III Reverse Transcriptase) was added to the cell lysate and incubated consecutively at 35 °C for 5 min, 45 °C for 20 min, 70 °C for 10 min, and 4 °C for chilling. For primer digestion, 5 µL of the primer digestion mix (1×ExoI Buffer, 7.5 U Exonuclease I) was prepared and incubated at 37 °C for 30 min, 80 °C for 20 min, and 4 °C for chilling. For the poly-A tailing reaction, 5 µL of the poly-A tailing reaction mix (4×PCR buffer, 5 mM dATP, 0.1 U RNase H, 60 U Terminal transferase) was prepared and incubated at 37 °C for 50 s, 65 °C for 10 min, and 4 °C for chilling. For the 2nd strand cDNA synthesis, 25 µL of the amplification mix (1×KAPA HiFi HotStart ReadyMix, 400 nM SMART_dT primer, 10 µL of poly-A tailing product, 1.5 µL of nuclease-free $H_2O$) was prepared and incubated at 95 °C for 3 min, 40 °C for 1 min, 65 °C for 10 min, and 72 °C for 5 min and chilled. For the cDNA amplification using an Ig-specific outer primer set, 20 µL of the outer amplification mix (1×KOD FX Neo buffer, 400 µM dNTPs, 1×UPM, 300 nM outer primer, 3.5 µL of the SMART product, 0.5 U KOD FX Neo polymerase) was prepared and incubated at 94 °C for 2 min, followed by 25 cycles at 98 °C for 10 s, 60 °C for 30 s, and 68 °C for 45 s, and then chilled. For cDNA amplification with an Ig-specific inner primer set, 20 µL of the inner amplification mix (1×KOD FX Neo buffer, 400 µM dNTPs, 500 nM 5' matrix primer mix, 500 nM 3' matrix primer mix, 3.5 µL of the outer cDNA amplified product, 0.5 U KOD FX Neo polymerase) was prepared and incubated at 94 °C for 2 min, followed by 25 cycles at 98 °C for 10 s, 60 °C for 30 s, and 68 °C for 45 s and then chilled. Finally, 3 µL of the amplified inner cDNA product was used for further plate indexing by amplification. Then, 20 µL of the plate-indexing amplification mix (1×KOD FX Neo buffer, 400 µM dNTPs; 500 nM i5-Fwd or i7-Fwd, 500 nM i7-Rev or i5-Rev; 2 µL of the inner cDNA amplified product; 0.5 U KOD FX Neo polymerase) was prepared and incubated at 94 °C for 2 min, followed by 25 cycles at 98 °C for 10 s, 60 °C for 30 s, and 68 °C for 45 s, and then chilled. All primers used are listed in *Supplementary file 1a and b*.

## Paired-end Illumina sequencing strategy and analysis

The libraries were sequenced on a MiSeq instrument (Illumina, San Diego, CA, USA) following the manufacturer's protocol. As the sequencing length was limited to 2×300 bp, we used the following strategy. Amplicons in both orientations were generated using an Illumina adapter sequence. Then, asymmetric 400+100 nt paired-end sequencing was performed to obtain high-quality 400+400 nt paired-end sequences of the original cDNA molecule. This approach provided a sufficient sequencing length to capture extra-long Ig variants. We followed the method by *Turchaninova et al., 2016*. Only reads that met the following four conditions were used as the resulting Ig sequences: (1) reads with a length of 260 nt or more; (2) cumulative clustered reads representing the top 5% of the entire reads; (3) 5'- and 3'-reads overlapping a length of 20 bases or more; and (4) a total length of 460±50 bp for heavy chain or 390±40 bp for light chain. All resulting sequence data were analyzed using IMGT/High V-QUEST (http://www.imgt.org/IMGT_vquest/vquest) for annotation (*Alamyar et al., 2012*). Using the sequences and annotations, Ig database construction and visualization of the Ig repertoire were performed using the in-house BONSCI software.

## Construction of a mouse IgG/IgK expression plasmid

The BsaI restriction sites were inserted into the paired B-cell repertoire amplicon from a single cell using PCR. The destination vector contained the IL6 signal peptide, EF1a promoter, ccdB gene, and mouse IgK Fc region. The donor vector contained the mouse IgG1 Fc region, the *Venus* gene, and the EF1a promoter. The paired B-cell repertoire amplicons, destination vector, and donor vector containing BsaI restriction sites were assembled using our assembly method. For this step, 10 µL of the assembly mix (1×T4 DNA ligase buffer, 1×BSA, 1 U BsaI restriction enzyme, 40 U T4 DNA ligase,

100 ng heavy chain amplicon, 100 ng light chain amplicon, 100 ng destination vector, and 100 ng donor vector) was prepared and incubated for 25 cycles at 37 °C for 3 min, 16 °C for 4 min, 50 °C for 5 min, and 80 °C for 5 min, and then chilled. The antibody sequence that entered the construct was fused to the *Venus* sequence and expressed in membrane form.

## Antibody display on FreeStyle 293 cultured cells

One microgram of antibody-expressing plasmid was transfected into $1\times10^6$ FreeStyle 293 cells in 1 mL culture using the 293fectin Transfection Reagent (Thermo Fisher Scientific) according to the manufacturer's instructions and cultured in FreeStyle 293 Expression Medium (Thermo Fisher Scientific) in a humidified incubator with 8% $CO_2$ at 37 °C and 125 rpm. Antibodies were displayed on the surface of cultured cells, which enabled the determination of antigen specificity. Furthermore, the antibody-display cells were tested for binding activity with Alexa647-labeled H1 and Alexa568-labeled H2 (Alexa Fluor Antibody Labeling Kits, ThermoFisher Scientific) using a BD FACSAria III (BD Biosciences).

## Antibody production

To obtain a large amount of secretory antibodies, PCR fragments from the antibody-expressing plasmid were cloned into the pcDNA3.4-mIgG1 or pcDNA3.4-kappa vectors. A mixture of 15 µg of pcDNA3.4-V-gene-mIgG1 vector and 15 µg of pcDNA3.4-V-gene-kappa vector was transfected into Expi293 cells using the ExpiFectamine 293 Transfection Kit (Thermo Fisher Scientific), according to the manufacturer's instructions, and cultured in Expi293 Expression Medium (Thermo Fisher Scientific) in a humidified 8% $CO_2$ incubator at 37 °C and 125 rpm. Then, 5 days post-transfection, the culture supernatants were harvested, and all proteins were purified with PureSpeed IMAC resin (Mettler Toledo, Columbus, OH, USA), according to the manufacturer's instructions.

## Surface plasmon resonance

Kinetic analyses were performed at 25 °C using a BIAcore 3000 machine (GE Healthcare Technologies). A2p1, B10p2, C10p2, E11p2, G6p2, A6p4, D4p4, D11p4, or G12p4 antibodies were immobilized on a CM5 sensor chip (GE Healthcare Technologies) with an amine-coupling kit according to the manufacturer's instructions. HA probes were serially diluted at five different concentrations and injected at a flow rate of 30 µL/min for 3 min with a dissociation time of 7 min in HBS–EP buffer (10 mM HEPES, pH 7.4, 150 mM NaCl, 3.4 mM EDTA, and 0.005% Surfactant P20). The chip was regenerated using 10 mM glycine (pH 2.5) as an HA probe.

## Acknowledgements

We thank Mr. Atsuo Kobayashi, Ms. Chieko Okamura, Dr. Seok-Won Kim, Mr. Maxime Hebrard, Dr. Todd D Taylor, and the RIKEN IMS Genome Platform for their technical support, Dr. Atsushi Miyawaki for gifting the Venus gene, and Dr. Christopher Nicholas for critical reading and English editing of the manuscript. We would like to thank Editage (https://www.editage.com) for English language editing. This study was supported in part by research grants from the RIKEN Research Program 'Single cell Project' (OO, TW, TK, and HF) and Grants-in-Aid for Scientific Research (S) [26221306 to TK and HF]. The funding sources were not involved in the study design, data collection and interpretation, or paper submission decisions.

## Additional information

### Funding

| Funder | Grant reference number | Author |
| --- | --- | --- |
| Japan Society for the Promotion of Science | 26221306 | Tomohiro Kurosaki Hidehiro Fukuyama |
| RIKEN | Single cell Project | Takashi Watanabe Tomohiro Kurosaki Osamu Ohara Hidehiro Fukuyama |

| Funder | Grant reference number | Author |
|---|---|---|

The funders had no role in study design, data collection and interpretation, or the decision to submit the work for publication.

## Author contributions
Takashi Watanabe, Hidehiro Fukuyama, Conceptualization, Resources, Funding acquisition, Validation, Visualization, Methodology, Writing - original draft, Project administration, Writing - review and editing; Hikaru Hata, Fumie Yokoyama, Tomoko Hasegawa, Investigation; Yoshiki Mochizuki, Naveen Kumar, Data curation, Software, Formal analysis; Tomohiro Kurosaki, Osamu Ohara, Supervision, Funding acquisition

## Author ORCIDs
Takashi Watanabe https://orcid.org/0000-0002-6184-0375
Osamu Ohara https://orcid.org/0000-0002-3328-9571
Hidehiro Fukuyama https://orcid.org/0000-0002-6457-0630

## Ethics
All animal experiments were performed using protocols approved by the Institutional Animal Care and Use Committee (IACUC) of the RIKEN Yokohama Branch (Project title: Immunological memory and vaccine development/Approval No.: 2019-001).

Reviewer #1 (Public review): https://doi.org/10.7554/eLife.95346.3.sa1
Reviewer #2 (Public review): https://doi.org/10.7554/eLife.95346.3.sa2
Author response https://doi.org/10.7554/eLife.95346.3.sa3

# Additional files

## Supplementary files
• Supplementary file 1. Supplementary tables. (a) Oligonucleotides. (b) Oligonucleotides.
• MDAR checklist

## Data availability
All raw and processed sequencing data generated in this study have been submitted to the NCBI Gene Expression Omnibus (GEO) under accession number GSE140720.

The following dataset was generated:

| Author(s) | Year | Dataset title | Dataset URL | Database and Identifier |
|---|---|---|---|---|
| Watanabe T, Ohara O | 2021 | Immunization with two heterotypic influenza hemagglutinins by time difference effectively produces broadly binding antibodies against heterotypic influenza virus | http://www.ncbi.nlm.nih.gov/geo/query/acc.cgi?acc=GSE140720 | NCBI Gene Expression Omnibus, GSE140720 |

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
