## [Editor Report · eLife Assessment]

The **useful** studies described here are broadly applicable to all antibody discovery subfields, even though they are not a significant improvement over published methods. The findings are **incomplete** with respect to the methodology, since details that are crucial in order to repeat the experiments are lacking (such as a timestamp). They also do not take into account multiple recent papers that have tested similar strategies. These studies will be of interest to a specialized audience working on generating antibodies to infectious agents.

---

## [Referee Report · Reviewer #1 (Public review)]

Summary:

This paper by Watanabe et al described an expression system that can express the paired heavy and light chains of IgG antibodies from single cell B cells. In addition, they used FACS sorting for specific antigen to screen/select the specific populations for more targeted cloning of mAb genes. By staining with multiple antigens, they were able to zoom in to cross-reactive antibodies.

Strengths:

A highly efficient process which combines selection/screening with dua expression of both antibody chains. It is particularly suitable for isolation of cross-reactive antibodies against conserved epitopes of different antigens, such as surface proteins of related viruses.

Weaknesses:

(1) The overall writing is very difficult to follow and the authors need to work on significant re-writing

(2) The paper in its current form really lacks detail and it is not possible for readers to repeat or follow their methods. For example: (a) It is not clear whether the authors checked the serum to see if the mice were producing antibodies before they sacrificed them to harvest spleen/blood i.e. using ELISA? (b) How long after administration of the second dose were the mice sacrificed? (c) What cell types are taken for single B cell sorting? Splenocytes or PBMC? These are just some of the questions which need to be addressed.

(3) According to the authors, 77 clones were sorted from the PR8+ and H2+ double positive quadrant. It is surprising that after transfection and re-analysing of bulk antibody presenting EXPI cells on FACS from, only 13 clones (or 8 clones? - unclear) seemed to be truly cross reactive. If that is the case, the approach is not as efficient as the authors claimed.

The authors have adequately addressed the issues raised

---

## [Referee Report · Reviewer #2 (Public review)]

Summary:

Watanabe, Takashi et al. investigated the use of the Golden Gate dual-expression vector system to enhance the modern standard for rapid screening of recombinant monoclonal antibodies. The presented data builds upon modern techniques that currently use multiple expression vectors to express heavy and light chain pairs. In a single vector, they express the linked heavy and light chain variable genes with a membrane-bound Ig which allows for rapid and more affordable cell-based screening. The final validation of H1 and H2 strain influenza screening resulted in 81 "H1+", 48 "H2+", and 9 "cross" reactive clones. The kinetics of some of the soluble antibodies were tested via SPR and validated with a competitive inhibition with classical well-characterized neutralizing clones.

Strengths:

In this study, Watanabe, Takashi et al. further develop and refine the methodologies for the discovery of monoclonal antibodies. They elegantly merge newer technologies to speed up turnaround time and reduce the cost of antibody discovery. Their data supports the feasibility of their technique.

This study will have an impact on pandemic preparedness and antibody-based therapies.

Weaknesses:

Limitations of this new technique are as follows: there is a significant loss of cells during FACs, transfection and cloning efficiency are critical to success, and well-based systems limit the number of possible clones (as the author discussed in the conclusions).

---

## [Author Response]

The following is the authors’ response to the original reviews.

**Reviewer #1 (Public Review):**
(1) The overall writing is very difficult to follow and the authors need to work on significant re-writing.

Thank you for your comment. We have rewritten the text and asked an immunology expert, who is also a native English speaking editor, to review it.

(2) The paper in its current form really lacks detail and it is NOT possible for readers to repeat or follow their methods. For example: (a) It is not clear whether the authors checked the serum to see if the mice were producing antibodies before they sacrificed them to harvest spleen/blood i.e. using ELISA? (b) How long after administration of the second dose were the mice sacrificed? (c) What cell types are taken for single B cell sorting? Splenocytes or PBMC?

Thank you for your comment. We have revised the methodology section thoroughly to ensure that the readers can follow and replicate the method. Our responses to the specific examples raised are as follows:

a) We did not examine the serum titer after immunization. An increased serum titer, as determined by ELISA, does not always reflect the number of cross-reactive B cells because we expected the serum titer to consist of polyclonal antibodies, which are a mixture of PR8-reactive, H2-reactive, and cross-reactive clones. We thus anticipated that we would not obtain enough cross-reactive B cells after a series of immunizations. After comparing various immunization methods, including different adjuvants and immunization sites, using the readout of the number of cross-reactive B cells, we decided to adopt the immunization protocol presented in this paper.

b) We sacrificed the mice two weeks after the second immunization (see Supplementary Figure 5).

c) For this experiment, we used CD43 MACS B cells from the spleen purified with negatively charged beads (see Supplementary Figure 6).

(3) According to the authors, 77 clones were sorted from the PR8+ and H2+ double positive quadrant. It is surprising that after transfection and re-analyzing of bulk antibody presenting EXPI cells on FACS, only 13 clones (or 8 clones? - unclear) seemed to be truly cross-reactive. If that is the case, the approach is not as efficient as the authors claimed.

Thank you for your comment. To isolate high affinity antibodies, we gated the high fluorescent intensity population of cross-reactive B cells during Ig-expressing 293 cell sorting, as shown in Fig 2B, while we collected a wide intensity population of cross-reactive cells during splenocyte sorting. The narrow gating reduced the number of clones. We, however, cannot quantify how many clones we lost in the process, but we achieved a cloning efficiency exceeding 75%. To avoid any confusion, we have clarified this point by attaching additional supplementary figures (Supplementary Figures 5 and 6).

**Reviewer #2 (Public Review):**
(4) A His tagged antigen was used for immunization and H1-his was used in all assays. Either the removal of His specific clones needs to be done before selection, or a different tag needs to be used in the subsequent assays.

Thank you for your comment. As pointed out, the possibility of antibody generation in regions other than HA cannot be ruled out since the immunized antigen and the detection antigen were the same. However, as shown in Table 1, the cross-reactive antibodies obtained in this study exhibited characteristic binding abilities to each of the six types of HA. If these were antibodies recognizing His, they would bind to all six types of HA. This indicates that these cross-reactive antibodies were not His-specific clones.

We have incorporated information on this potential caveat into the discussion (page 12, lines 4-9).

(5) This assay doesn't directly test the neutralization of influenza but rather equates viral clearance to competitive inhibition. The results would be strengthened with the demonstration of a functional antibody in vivo with viral clearance.

Thank you for your constructive comment. While we agree that demonstration of a functional antibody in vivo with viral clearance would strengthen our results, this is clearly out of the scope of our current study and will be subject of future research.

(6) Limitations of this new technique are as follows: there is a significant loss of cells during FACs, transfection and cloning efficiency are critical to success, and well-based systems limit the number of possible clones (as the author discussed in the conclusions). Early enrichment of the B cells could improve efficiency, such as selection for memory B cells.

Thank you for your comment. Our cloning efficiency for sorted B cells exceeded 75%. However, we selected high binders of cross-reactive B cells during Ig-expressing 293 functional screening on purpose, as shown in Figure 2B, while we collected all cross-reactive B cells during B cell sorting (see attached Supplementary Figure 5). This functional selection step reduced the number of clones. We clarified this point by attaching additional supplementary figures (Supplementary Figures 5 and 6).

Our sorted cross-reactive B cells are most likely CD38+ memory B cells, as shown in Supplementary Figure 6.

**Reviewer #1 (Recommendations For The Authors):**
a) It is advised for the authors to provide a flow chart with time stamps to prove the many statements made in the paper. For example, it is stated that "we demonstrated efficient isolation of influenza cross-reactive antibodies with high affinity from mouse germinal B cells over 4 days". It is not clear how this was calculated.

Thank you for your comment. We have prepared a time-stamped flow chart (Supplementary Figure 5).

b) The papers cited by the authors are relatively old if not outdated. There are many papers published focusing on efficient isolation of mAbs for SARS-CoV-2 research. For example, the paper by Lima et al (Nat Comm 2022, 13:7733) used a very similar strategy for rapid isolation of cross-reactive mAbs by FACS sorting followed by cloning of paired heavy and light chains from single B cells. The authors need to incorporate citations from the latest publications in this field.

Thank you for your comment. The paper by Lima et al. (Nat Comm 2022, 13:7733) has been cited in the Discussion as ref 28.

c) Figure 2 needs much more detail for readers to follow.

Thank you for your comment. We have revised the legend of Figure 2 accordingly and added additional supplementary figures (Supplementary Figures 5 and 6) to increase clarity.